# Uncertainty Quantification of Ride Comfort Based on gPC Framework for a Fully Coupled Human–Vehicle Model

Byoung-Gyu Song [1,†], Jong-Jin Bae [2,†] and Namcheol Kang [1,*]

1   School of Mechanical Engineering, Kyungpook National University, Daegu 41566, Republic of Korea; sbg045@knu.ac.kr
2   Korea Aerospace Research Institute, KSLV-II R&D Directorate, Daejeon 34133, Republic of Korea; jjbae@kari.re.kr
*   Correspondence: nckang@knu.ac.kr
†   These authors contributed equally to this work.

**Abstract:** We investigated the stochastic response of a person sitting in a driving vehicle to quantify the impact of an uncertain parameter important in controlling defect reduction in terms of ride comfort. Using CarSim software and MATLAB/Simulink, we developed a fully coupled model that simulates a driving vehicle combined with an analytical nonlinear human model. Ride comfort was evaluated as a ride index considering the frequency weights defined in BS 6841. Additionally, to investigate the uncertainty of the ride index, a framework for calculating the ride index was proposed using the generalized polynomial (gPC) method. Further, sensitivity analysis of the ride index was performed for each uncertainty parameter, such as stiffness and damping. The results obtained through the gPC method were in good agreement with those obtained via Monte Carlo simulation (MCS) and were excellent in terms of computation time without a loss of numerical accuracy. Through in-depth investigation, we found that the stochastic distribution of the ride index varies differently for each uncertain parameter in the human model. By comparing linear and nonlinear human models, we also found that the nonlinearity of the human model is an important concern in the stochastic estimation of ride comfort.

**Keywords:** human–vehicle model; uncertainty; ride index; generalized polynomial chaos; stochastic analysis





## 1. Introduction

The dynamic properties of the human body have received attention in recent years due to the fact that the response of whole-body vibrations is closely related to the ride comfort of occupants in transportation systems. Meanwhile, a quantitative understanding of how a seated human body responds to whole-body vibrations allows automotive engineers to design more human-oriented vehicles. In addition, whole–body vibration has been recently considered in the design and assessment of road infrastructure (paved roads, highways, bridges, etc.) [1–5]. However, despite many efforts to predict ride comfort, the quantification of ride comfort remains challenging as it relies on the perceptions and emotions that are influenced by many factors, such as driving and environmental conditions.

In the automotive industry, both subjective and objective measurements are commonly used to evaluate the ride comfort of a vehicle. In subjective measures, various kinds of experimental studies have been used to correlate with subjective ratings of occupants exposed to specific driving or seating conditions [6,7]. On the other hand, the quantitative analysis of whole-body vibrations is attracting attention in order to objectively investigate ride comfort because this evaluation can detect small variations in the passenger's dynamic response compared to subjective evaluations [8]. This vibration method is widely used in various applications, such as conventional vehicles [9], railway vehicles [10], high-speed electric multiplexers [11], and helicopters [12]. Furthermore, the spread of electric or

hybrid vehicles has led to a new turning point in the study of vibration analysis in the automotive industry. In general, these vehicles do not mask road-induced vibrations in the low-frequency range owing to the absence of a combustion engine [13]. Therefore, understanding human responses to vibrations will become even more important in the near future.

Because experimental approaches for the evaluation of ride comfort are time-consuming and costly, increasing attention is being paid to the industrial requirements for computational analysis based on dynamic simulations. Brogioli et al. [14] developed a model that combines a seat and a human as well as a five-degrees-of-freedom mathematical model, computed weighted accelerations, and a Seat Effective Amplitude Transmissibility (SEAT) index. Additionally, Mohajer et al. [15,16] introduced a biomechanical human model obtained using a computational multibody system and investigated the ride comfort index with respect to road roughness and the longitudinal velocity of a vehicle. Anandan et al. [17] adopted a human–vehicle model with 14 degrees of freedom to analyze the ride comfort of the proposed active suspension vehicle system. Jun Wu et al. [18,19] developed a train–seat–human model with vertical, lateral, and roll vibrations to study the ride comfort of rail vehicles.

In general, the parameters of a real human body (e.g., weight, length, and stiffness) are highly uncertain compared to typical mechanical structures [20,21]. Furthermore, human parameters result in time-variant systems because of the variations in muscle tension, blood flow, and other environmental circumstances. Consequently, the dynamic response of the human body to whole-body vibrations is uncertain due to the uncertainty of the parameters. Therefore, numerical calculations of dynamic human responses to whole-body vibrations using deterministic parameters are limited in the sense that they do not provide a unique subjective rating of feeling.

Numerous stochastic methods have been introduced to calculate the response of a system in the presence of parameters with uncertainty. Among them, Monte Carlo simulation (MCS) is widely used to estimate system uncertainty [22–26]. In MCS, a set of samples of an input parameter with uncertainty are randomly selected from the corresponding probability distributions that are estimated or assumed to be given, and multiple simulations are taken to compute the corresponding output for each set of samples. However, due to the nature of MSC and random sampling, a huge number of simulations are required, which can be computationally time-consuming. In order to reduce the computational cost, many studies have proposed to use the quasi-Monte-Carlo method with an improved sampling algorithm [27,28]. Despite these efforts, stochastic methods based on MCS can be computationally time-consuming as they typically require a large number of simulations. To deal with modeling uncertainty more efficiently, polynomial chaos expansion has become an important method [29,30], which is based on the homogeneous chaos theory of Wiener [31]. The polynomial chaos method has been applied to various engineering fields, such as vehicle dynamics [32–36].

In this study, we analyzed the ride comfort stochastically to quantify the effect of uncertain human parameters. To calculate the ride comfort of a seated person in a driving vehicle on a rough road, we developed a fully coupled human–vehicle model using a vehicle in CarSim combined with a nonlinear human model in MATLAB/Simulink. The ride comfort was quantified as s ride index according to BS 6841 [37], which considers frequency-weighted accelerations. For stochastic calculations, we adopted an approximate model based on gPC (generalized polynomial chaos) and compared it with MSC. In order to investigate the effect of uncertainty, the stochastic distributions of the ride index were calculated for each uncertain parameter of the human model, such as stiffness and damping. Furthermore, we found significant differences between linear and nonlinear human models in terms of uncertain parameters.

## 2. Human–Vehicle Model and Road Profile for Stochastic Analysis

This section describes a human–vehicle model, road surface generation, and an approximate model based on gPC. The coupling between the human body and the vehicle model is implemented in MATLAB/Simulink. The road profile is generated using the road roughness classification method based on power spectral density. Finally, an approximate model based on generalized polynomial chaos expansion (gPC) is developed for stochastic analysis [29].

### 2.1. Model Description and Derivation of Equation of Motion

To calculate the responses of the seated human, the nonlinear five-degree-of-freedom human model, which was used in the previous works [38], was considered in this study. The schematic of the proposed human model is illustrated in Figure 1. The lumped parameter of the human model is composed of a head, a trunk, and a thigh with mass and moment of inertia. Each segment is connected using a torsional spring and a damper, and the trunk and thigh contact the seat through a spring and dampers, which are effective parameters combined with seat cushion and human skin. Regarding the generalized coordinates of the hip joint, $x_h$ and $z_h$ indicate the horizontal and vertical displacement of the hip joint, respectively, and $\theta_1$, $\theta_2$, and $\theta_3$ denote the angular displacement of the thigh, trunk, and head, respectively. Only a normal deflection of the springs and dampers is considered for the calculation of the spring and damping forces, respectively (see Figure 2). In fact, the effective stiffness, $k_i$, connected to the thigh has different values depending on the contact location according to the experimental results [39]. However, due to the fact that the effect of the uncertainty of the design parameters on human–vehicle ride comfort is the main interest of our study, we adopted an identical value for the simplicity of analysis.

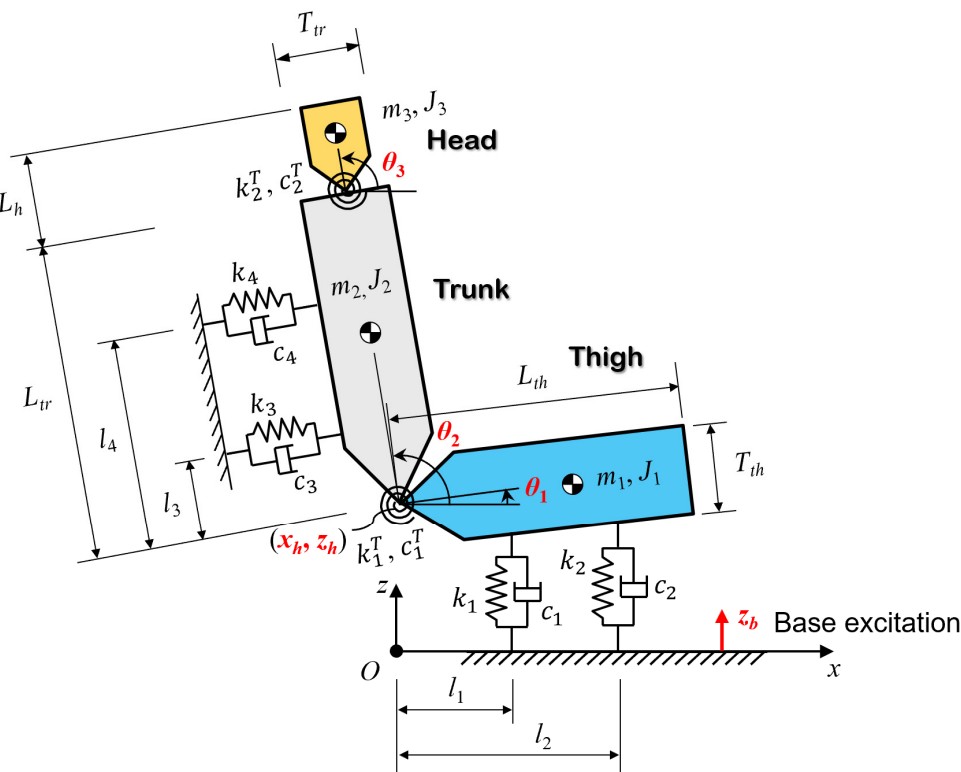

**Figure 1.** The schematic diagram of the five-degree-of-freedom human model.

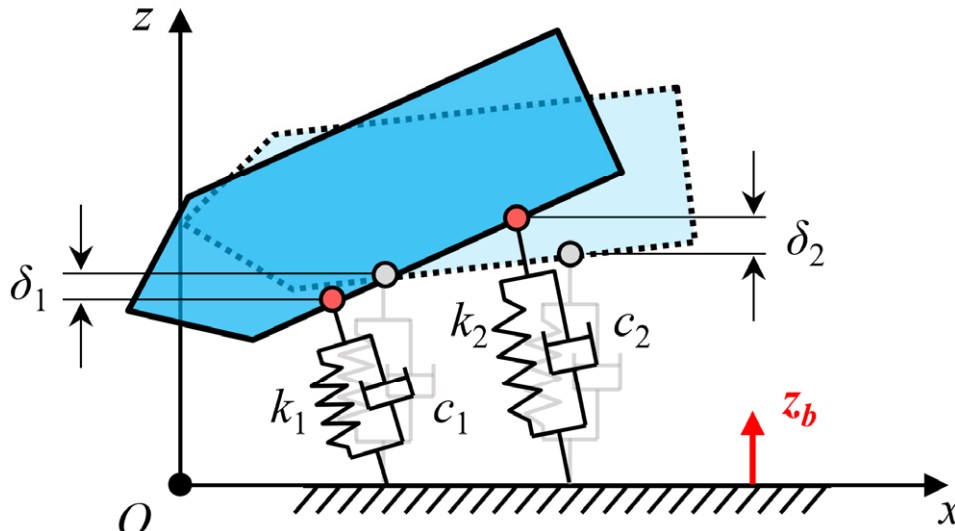

**Figure 2.** Geometric sketch to represent the deflections of translational springs.

The nonlinear equation of the five-degree-of-freedom human model is derived using Lagrange's equation. Kinetic energy ($T$), potential energy ($V$), and Rayleigh's dissipation ($D$) of each segment are derived based on the displacement of the center of gravity, as follows:

$$T = \frac{1}{2}\sum_{i=1}^{3} m_i\left(\dot{x}_i^2 + \dot{z}_i^2\right) + \frac{1}{2}\sum_{i=1}^{3} J_i\dot{\theta}_i^2$$

$$V = \frac{1}{2}\sum_{i=1}^{4} k_i\delta_i^2 + \frac{1}{2}\sum_{i=1}^{2} k_i^T\left(\theta_i^T\right)^2 + \sum_{i=1}^{3} m_i g\Delta_i \qquad (1)$$

$$D = \frac{1}{2}\sum_{i=1}^{4} c_i\dot{\delta}_i^2 + \frac{1}{2}\sum_{i=1}^{2} c_i^T\left(\dot{\theta}_i^T\right)^2$$

where $m_i$ and $J_i$ denote the mass and mass moment of inertia of each segment, respectively. Additionally, $k_i$ and $c_i$ are translational, and $k_i^T$ and $c_i^T$ are torsional stiffness and damping coefficients, respectively. The translational and angular displacements of the springs are represented by $\delta_i$ and $\theta_i^T$, respectively. In addition, $g$ is the gravitational acceleration and $\Delta_i$ is the vertical displacement of each segment. The positions of the center of gravity at each segment are as follows:

$$x_1 = x_h + \tfrac{1}{2}L_{th}\cos\theta_1,$$
$$x_2 = x_h + \tfrac{1}{2}L_{tr}\cos\theta_2,$$
$$x_3 = x_h + L_{tr}\cos\theta_2 + \tfrac{1}{2}L_h\cos\theta_3,$$
$$z_1 = z_h + \tfrac{1}{2}L_{th}\sin\theta_1,$$
$$z_2 = z_h + \tfrac{1}{2}L_{tr}\sin\theta_2,$$
$$z_3 = z_h + L_{tr}\sin\theta_2 + \tfrac{1}{2}L_h\sin\theta_3$$

where $x_h$ and $z_h$ are the horizontal and vertical displacement of the hip joint. The length of the thigh, trunk, and head are denoted as $L_{th}$, $L_{tr}$, and $L_h$, respectively. Accordingly, the translational and angular displacements of each spring are given below.

$$\delta_1 = z_h - \tfrac{1}{2}T_{th}\cos\theta_1 + L_1\sin\theta_1 - z_b - \delta_1^*$$
$$\delta_2 = z_h - \tfrac{1}{2}T_{th}\cos\theta_1 + L_2\sin\theta_1 - z_b - \delta_2^*$$
$$\delta_3 = x_h\sin\theta_2^* - z_h\cos\theta_2^* - L_3\sin(\theta_2 - \theta_2^*) - \tfrac{1}{2}T_{tr}\cos(\theta_2 - \theta_2^*) - x_3^*\sin\theta_2^* + z_3^*\cos\theta_2^* - \delta_3^*$$
$$\delta_4 = x_h\sin\theta_2^* - z_h\cos\theta_2^* - L_4\sin(\theta_2 - \theta_2^*) - \tfrac{1}{2}T_{tr}\cos(\theta_2 - \theta_2^*) - x_4^*\sin\theta_2^* + z_4^*\cos\theta_2^* - \delta_4^*$$
$$\theta_1^T = \theta_2 - \theta_1 - \theta_1^*$$
$$\theta_2^T = \theta_3 - \theta_2 - \theta_2^*$$

where the thickness of the thigh and trunk are denoted as $T_{th}$ and $T_{tr}$, respectively. In addition, $\delta_i^*$ and $\theta_i^*$ indicate the initial length and angle of the translational and torsional springs, respectively, and $(x_i^*, z_i^*)$ indicate the position where the springs are fixed at backrest. The inertial, geometric values, stiffness, and damping coefficients used in this study are listed in Appendix A. Now applying Lagrange's equation using Equation (1) yields

$$
\begin{bmatrix}
S_{11} & 0 & S_{13} & S_{14} & S_{15} \\
 & S_{22} & S_{23} & S_{24} & S_{25} \\
 & & S_{33} & 0 & 0 \\
 & Sym. & & S_{44} & S_{45} \\
 & & & & S_{55}
\end{bmatrix}
\begin{pmatrix}
\ddot{x}_h \\
\ddot{z}_h \\
\ddot{\theta}_1 \\
\ddot{\theta}_2 \\
\ddot{\theta}_3
\end{pmatrix}
=
\begin{pmatrix}
P_1 \\
P_2 \\
P_3 \\
P_4 \\
P_5
\end{pmatrix}
\tag{2}
$$

The elements of each matrix are given in Appendix B.

### 2.2. Random Road Profile and Human–Vehicle Model

In order to analyze the effect of road surface on human–vehicle vibrations, we generated random road profiles according to ISO 8608, which classifies road roughness using power spectral density (PSD) [40]. An approximate form of the road displacement PSD can be written as follows:

$$
\Phi(\Omega) = \Phi(\Omega_0)\left(\frac{\Omega}{\Omega_0}\right)^{-w}
\tag{3}
$$

where $\Omega$ is the angular spatial frequency and $w$ is the waviness. The reference spatial frequency of $\Omega_0$ is given by 1 rad/m. In addition, the PSD function can be modified as follows [41,42].

$$
\Phi(\Omega) = \frac{2\alpha\sigma^2}{\sigma^2 + \alpha^2}
\tag{4}
$$

where $\sigma^2$ is the variance in the road roughness, and $\alpha$ is the coefficient with respect to the type of road surface. The road roughness variances and coefficients are listed in Table 1. Additionally, the road profile can be approximated using the superposition of sinusoidal functions with respect to longitudinal distance, as follows:

$$
z(x) = \sum_{i=1}^{N} A_i \sin(\Omega_i x - \phi_i)
\tag{5}
$$

where the random phase angle is uniformly distributed in the range of $0 \leq \phi_i < 2\pi$, and $A_i$ denotes the amplitudes of the road profiles, defined as follows:

$$
A_i = \sqrt{\Phi(\Omega_N)\frac{\Delta\Omega}{\pi}}
\tag{6}
$$

where

$$
\Delta\Omega = \frac{\Omega_N - \Omega_1}{N - 1}
$$

**Table 1.** Standard deviation of the road roughness with respect to the road class at $\Omega_0 = 1$ rad/m [42].

| Road Class | $\sigma$ ($10^{-3}$ m) | $\Phi(\Omega_0)$ ($10^{-6}$ m³) | $\alpha$ (rad/m) |
|---|---|---|---|
| A (very good) | 2 | 1 | 0.127 |
| B (good) | 4 | 4 | 0.127 |
| C (average) | 8 | 16 | 0.127 |
| D (poor) | 16 | 64 | 0.127 |
| E (very poor) | 32 | 256 | 0.127 |

In this study, $\Omega_N = 10\pi$ (rad/m) and $\Omega_1 = 0.02\pi$ (rad/m) are considered to generate a random road profile that covers the frequency range of the weighting functions in BS

6841. We generated the power spectral densities of five road profiles and compared them with eight road roughness references, as shown in Figure 3. Note that the higher the power spectral density, the rougher the road, and the resulting road profile falls into the ISO 8608 classification category. In addition, Figure 4 shows the road profiles generated from the power spectral densities for five different road roughness.

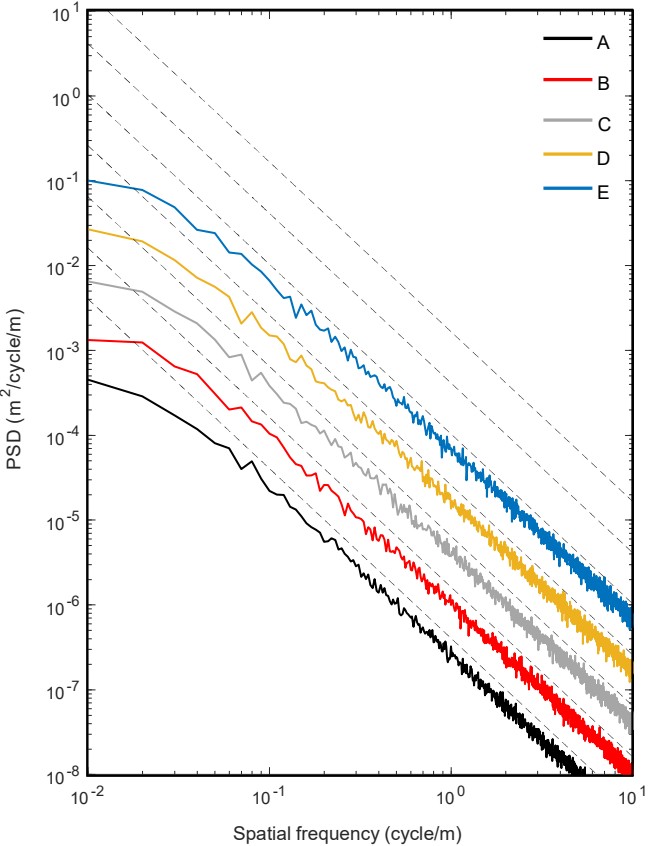

**Figure 3.** Power spectral densities of the generated road profiles according to ISO 8608 (gray dotted line: reference values of the road classification).

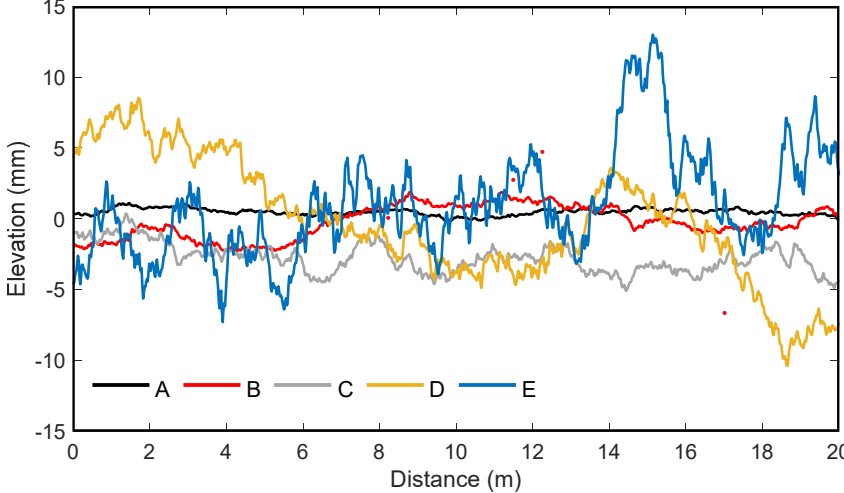

**Figure 4.** Road profiles with respect to longitudinal distance for five road roughness.

As shown in Figure 5, to simulate the human–vehicle model, the mathematical human model is fully coupled with the vehicle model using CarSim 9.0.2 [43] in MATLAB/Simulink. The vibrations caused by the vehicle movement are transmitted to the

human body model as a base excitation caused by the displacement of the vehicle seat floor. Additionally, the reaction forces caused by the human body are transmitted to the seat floor of the vehicle through the springs and dampers. The spring and damping forces are calculated considering the displacements between the thigh and the seat pad of the human model, as follows:

$$F_R = \sum_{i=1}^{4} k_i \delta_i + \sum_{i=1}^{4} c_i \dot{\delta}_i \tag{7}$$

where $\delta_i$ and $\dot{\delta}_i$ indicate the translational displacements and their time rate of changes of each spring, respectively.

**Figure 5.** Schematic of full coupling analysis for human–vehicle simulation.

Figure 6 illustrates the effects of road roughness and coupling on the reaction forces and the vertical displacement of the seat floor. Figure 6a shows the vertical forces generated between the human model and the vehicle model in the case of full coupling analysis. The average reaction force corresponding to the weight of the human model is about 500 N, regardless of the road class, either A or E. However, the reaction force in the E class, which is classified as a rough road, shows greater fluctuation than when the A class is used, which is classified as a smooth road. Its magnitude is more than 20% greater than average. In addition, Figure 6b shows the vertical displacement of the driver's seat when the human–vehicle model is connected with full coupling and partial coupling on an ISO E-class road. Full coupling means that the human–vehicle model transmits reaction forces in the mutual direction, and partial coupling means that only the vehicle transmits reaction forces to the human model. The vehicle displacement in the case of full coupling is smaller than the partial coupling case because it ignores the weight of the human model. Although the displacement difference between the full and partial couplings is not significant, coupling analysis is considered in this study because the accuracy of the simulation is crucial to the analysis of the standard deviation when the uncertainty of the human parameters is included.

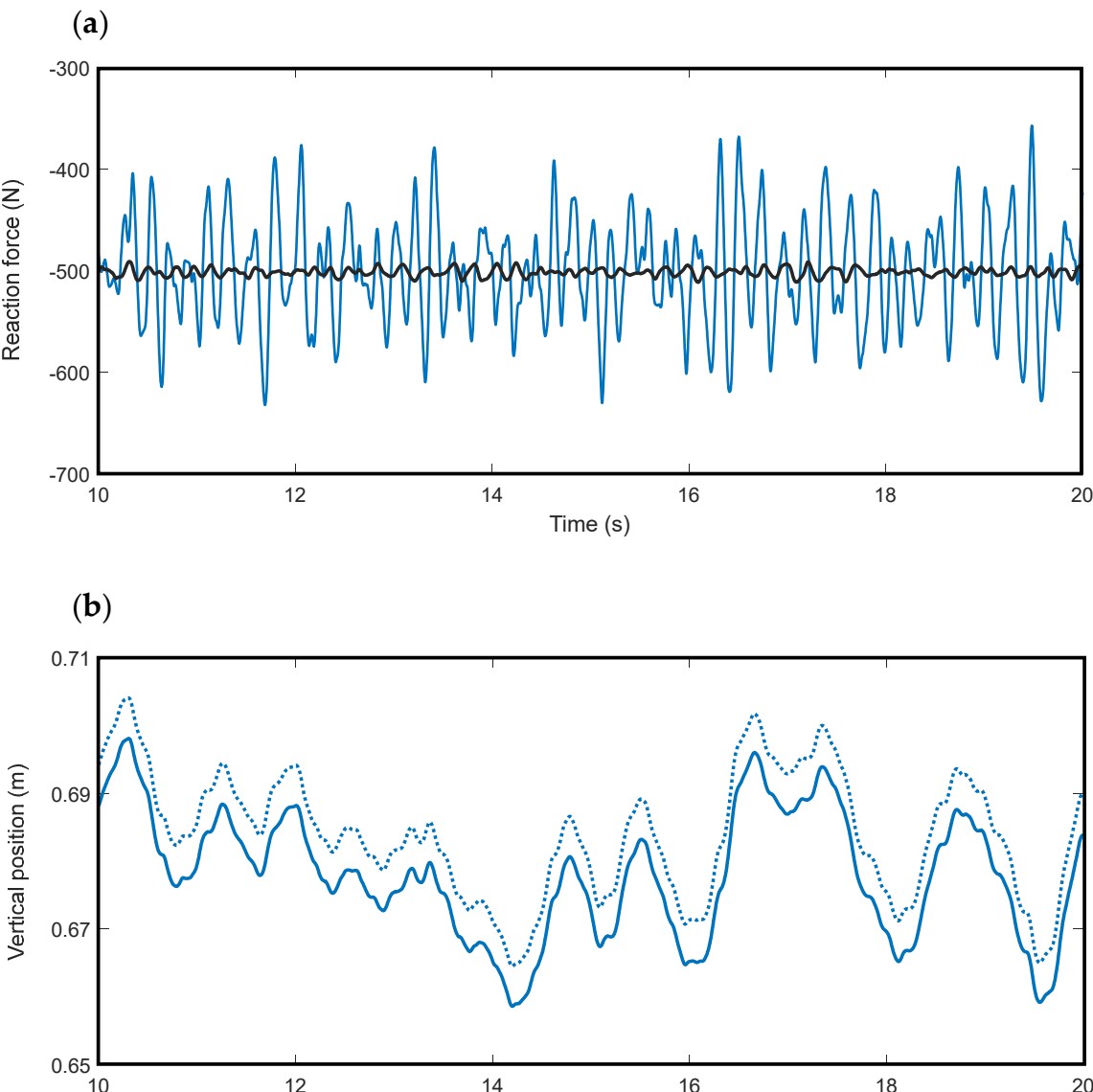

**Figure 6.** The effects of road roughness and coupling on the reaction forces and vertical displacement of the seat floor. (**a**) Reaction force in A- and E-class roads. (**b**) Vertical displacement for the case of full and partial coupling analysis on an E-class road.

### 2.3. Approximate Model Based on Generalized Polynomial Chaos

Suppose the uncertain parameter of the stochastic model is expressed in a set of random variables, $\boldsymbol{\xi} = (\xi_1, \xi_2, \ldots, \xi_n)$, the model response can also be expressed using the same set of variables [44]. Therefore, the output $\widetilde{y}(t, \boldsymbol{\xi})$ of the stochastic model can be approximated using truncated generalized polynomial chaos expansion (gPC) expressed as the set of $\boldsymbol{\xi}$, as given by [29,45]

$$\widetilde{y}(t, \boldsymbol{\xi}) = \sum_{j=0}^{S} a_j(t) \phi_j(\boldsymbol{\xi}) \tag{8}$$

where $\phi_j(\boldsymbol{\xi})$ are the generalized Askey–Wiener polynomial chaos basis functions in terms of multi-dimensional random variables $\boldsymbol{\xi} = (\xi_1, \xi_2, \ldots, \xi_n)$. Additionally, $a_j$ are the deter-

ministic coefficients to be estimated. Further, the total number of polynomial terms, $S$, can be obtained as follows:

$$S = \frac{(p+n)!}{p!n!} - 1 \tag{9}$$

here, $p$ indicates the order of the polynomial chaos, and $n$ indicates the number of uncertain parameters. The orthogonal polynomial functions according to the Askey–Wiener polynomial basis are listed in Table 2.

**Table 2.** Distributions of random variables and corresponding polynomial basis function.

| Distribution | Polynomial Function | Support |
|---|---|---|
| Gaussian | Hermite | $(-\infty, +\infty)$ |
| Gamma | Laguerre | $[0, +\infty)$ |
| Beta | Jacobi | $[a, b]$ |
| Uniform | Legendre | $[a, b]$ |

In this study, we assumed that the uncertain parameter has a Gaussian distribution. For the random variable with a Gaussian distribution, generalized polynomial chaos expansion employs the Hermite polynomials. The general expression of Hermite polynomials of order $p$ is expressed as follows:

$$\phi_p(\xi_1, \xi_2, \ldots, \xi_p) = (-1)^p e^{\frac{1}{2}\xi^T \xi} \cdot \frac{\partial^p}{\partial \xi_1 \cdot \partial \xi_2 \cdots \partial \xi_p} \cdot e^{-\frac{1}{2}\xi^T \xi} \tag{10}$$

where superscript $^T$ indicates the transpose of the random variable vector.

The unknown coefficients of polynomial chaos expansion are commonly obtained through the use of the Galerkin method. However, it is not easy to apply the Galerkin method when the stochastic model equation is complicated and difficult to tractable. Therefore, in this study, the unknown coefficient was estimated using the collocation method. The method is accomplished by computing the output of the stochastic model at a set of collocation points. The collocation points of the random variable are the roots of the next higher-order polynomial [36,44]. For example, in a one-dimensional random field, the collocation points of the second-order polynomial chaos are calculated from the root of the third-order Hermite polynomial $\left(-\sqrt{3}, 0, \sqrt{3}\right)$. In the collocation method, the output of the stochastic model and approximation model based on gPC are the same at the collocation point, as follows:

$$y(t, \xi) = \widetilde{y}(t, \xi) = \mathbf{T}(\xi)a(t) \tag{11}$$

where $y(t, \xi)$ represent the output vectors of the stochastic model at the collocation points, $\xi = (\xi_1, \xi_2, \cdots, \xi_M)$. The subscript of $M$ denotes the number of collocation points. The transform matrix $\mathbf{T}$ and unknown coefficients $a(t)$ are as follows:

$$\mathbf{T}(\xi) = \begin{pmatrix} \phi_0(\xi_1) & \phi_1(\xi_1) & \cdots & \phi_S(\xi_1) \\ \phi_0(\xi_2) & \phi_1(\xi_2) & \cdots & \phi_S(\xi_2) \\ \vdots & \vdots & \ddots & \vdots \\ \phi_0(\xi_M) & \phi_1(\xi_M) & \cdots & \phi_S(\xi_M) \end{pmatrix}, \; a(t) = \begin{pmatrix} a_1(t) \\ a_2(t) \\ \vdots \\ a_M(t) \end{pmatrix}$$

Applying the least square method to Equation (10), the vectors of unknown coefficient can be easily obtained, as follows:

$$a(t) = \left\{ \mathbf{T}(\xi)^T \mathbf{T}(\xi) \right\}^{-1} \mathbf{T}(\xi)^T y(t, \xi) \tag{12}$$

Using the calculated coefficients of the orthogonal polynomials, the mean and variance can be obtained, as follows:

$$\mu = a_0 \phi_0(\xi) \text{ and } \sigma^2 = \sum_{j=1}^{S} a_j^2 \left\langle \phi_j^2 \right\rangle \tag{13}$$

where $\langle \cdot, \cdot \rangle$ denotes the ensemble average inner product [29].

We used an approximate model based on gPC to analyze the stochastic results of the human–vehicle model instead of Monte Carlo simulation (MCS), which requires a large number of simulations. The proposed approximate model consists of four major steps. First, the selection of the uncertain parameters according to an assumed distribution. Second, the generation of an approximate model through generalized polynomial chaos expansion. Third, the estimation of the coefficients using the collocation method, and finally, a comparison of the stochastic results obtained through gPC with those obtained through MCS. This process of the calculation is illustrated in Figure 7. In essence, gPC can compute the stochastic results of a dynamic system using the polynomials, as opposed to fully integrating the equations of motion through the use of MCS. Therefore, the computational time of gPC is significantly reduced compared to that of MCS. Essentially, in contrast to fully integrating the equations of motion through MCS, gPC can use polynomials to compute the stochastic results of a dynamic system. Therefore, the computational time of gPC is significantly decreased compared to MCS.

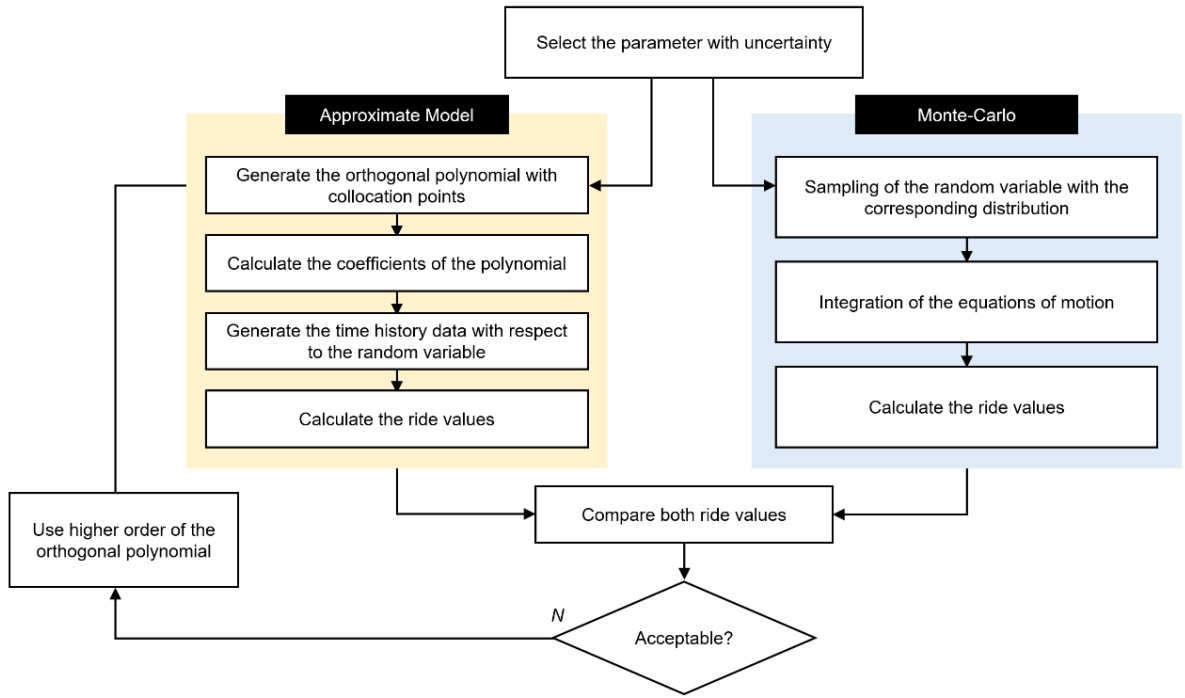

**Figure 7.** Flowchart of the approximate model based on gPC and Monte Carlo simulation.

## 3. Stochastic Analysis of Ride Comfort Index

In this section, we analyze the effect of uncertain parameters on ride comfort. We use the generalized polynomial chaos (gPC) method to obtain the stochastic responses of the human model. The gPC results are compared with Monte Carlo simulations (MCS) in terms of accuracy and time efficiency. Finally, a parametric study of stochastic ride comfort is performed.

### 3.1. gPC Results for Human–Vehicle Model with Uncertain Parameters

A fully coupled human–vehicle simulation was performed by combining CarSim and MATLAB/Simulink in the presence of the parametric uncertainty of the human body model. In this stochastic analysis, it is assumed that there are three uncertain parameters, such as two translational stiffness ($k_1$, $k_3$) and one torsional stiffness ($k_1^T$). It also assumes that the translational springs connected to the same human body are identical to reduce computational costs, that is, $k_1 = k_2$ and $k_3 = k_4$ (see Figure 1). A single uncertain parameter was selected for the dynamic simulations in turn, and the accelerations of the human model were extracted with respect to the corresponding parameter. The second order of the polynomial chaos was considered to obtain the stochastic responses of the human model. In other words, single-dimensional second-order polynomial chaos expansion was used in the approximate model based on gPC. In addition, those with uncertain stiffness were assumed to be the Gaussian distributed variable. They can be represented as follows:

$$
\begin{aligned}
k_1(\xi_1) &= \mu_{k_1} + \sigma_{k_1}\xi_1 \\
k_3(\xi_2) &= \mu_{k_3} + \sigma_{k_3}\xi_2 \\
k_1^T(\xi_3) &= \mu_{k_1^T} + \sigma_{k_1^T}\xi_3
\end{aligned}
\tag{14}
$$

where $\mu_k$ and $\sigma_k$ represent the mean and standard deviation values of the corresponding stiffness, respectively. In this study, the standard deviation of the uncertain stiffness was assumed to be 5% of the corresponding mean value. The conditions of the human–vehicle simulations are as follows. The simulated vehicle model used in CarSim was an E-class sedan. In each segment, the accelerations of the human model were calculated when the vehicle was traveling at a constant speed of 60 km/h in the longitudinal direction. The steering angle of the vehicle was controlled to maintain straight-line driving. The total running time and sampling frequency were set to 50 s and 1000 Hz, respectively. However, in the dynamic simulations, only times between 5 and 50 s are analyzed to eliminate transient effects due to accelerations fluctuating to the target speed.

Since MCS requires a huge number of simulations to achieve satisfactory convergence, we also developed an automatic process for human–vehicle simulations using the COM interface in CarSim. Automation is implemented by generating a set of samples for uncertain parameters and controlling the COM interface with a MATLAB script. A detailed description of the automatic process follows. First, generate the samples corresponding to the probability distribution in the MATLAB script. The number of samples at this stage depends on the number of MCS and gPC runs. Second, launch MATLAB/Simulink in CarSim after executing the COM interface of CarSim in the MATLAB script. Third, provide the generated set of samples as input to the function corresponding to the human model in MATLAB/Simulink. Fourth, once the simulation is performed, save the response of generalized coordinates of the human model for each sample in the workspace of MATLAB. Then, terminate the CarSim and MATLAB/Simulink via MATLAB script. Finally, repeat steps 2 to 4 as many times as the number of simulations of MCS and gPC.

For the three uncertain parameters $k_1$, $k_3$, and $k_1^T$, the variations of the vertical acceleration at the hip joint of the human model were calculated using gPC and MCS, as shown in Figure 8. Vertical accelerations close to zero are displayed in light gray to indicate the distribution of probability densities (see the color bar on the right). Several results are observed. First, it is clear that the accelerations calculated by the approximation model based on gPC show good agreement with those obtained through the use of MCS for all three types of uncertain parameters. To obtain a satisfactory convergent result, gPC requires only three times simulations for three collocation points, while MCS does 500 times simulations. The former is at least 160 times more efficient than the latter. Second, although the average values of the acceleration are the same, the distributions of the probability densities are different depending on the uncertain parameters. Third, when $k_1^T$ is uncertain in Figure 8e,f, the probability density of acceleration appears as a solid line instead of a

distribution due to a small standard deviation. This is because that $k_1^T$ is located at the very position of the hip joint that we calculated.

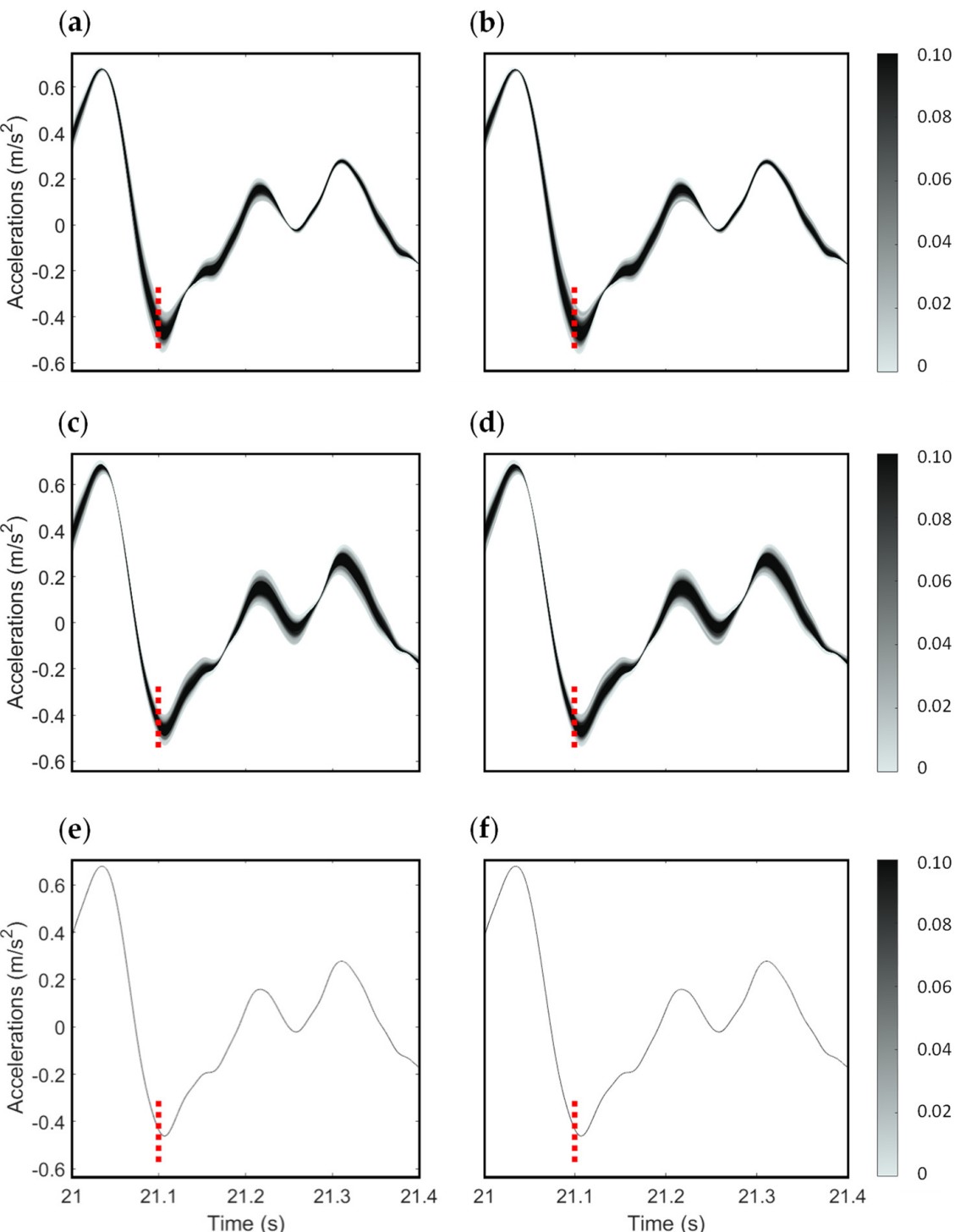

**Figure 8.** Variations in the vertical acceleration at the hip joint for several uncertain parameters. Subfigures in (**a**,**b**), (**c**,**d**), and (**e**,**f**) are for uncertain parameters $k_1$, $k_3$, and $k_1^T$, respectively. Additionally, figures in (**a**,**c**,**e**) are calculated using gPC, and in (**b**,**d**,**f**) are calculated using MCS.

In order to investigate the effect of the variations on vertical accelerations, we calculated the probability densities for the uncertain parameter $k_1$ and $k_3$ in detail. For example, the probability density function of the acceleration at $t$ = 21.1 s (red dashed line in Figure 8)

is shown in Figure 9. Here, the shaded blocks and solid lines represent the MCS and gPC results, respectively. In the case of uncertain parameter $k_1$, a wider distribution is observed than that of $k_3$. From a design point of view, it is crucial that the variation of the acceleration shows a different distribution depending on the uncertainty of the parameter. However, it is not practical to examine the variations for all time responses. Therefore, it is necessary to quantify a specific value that takes into account all of the time responses.

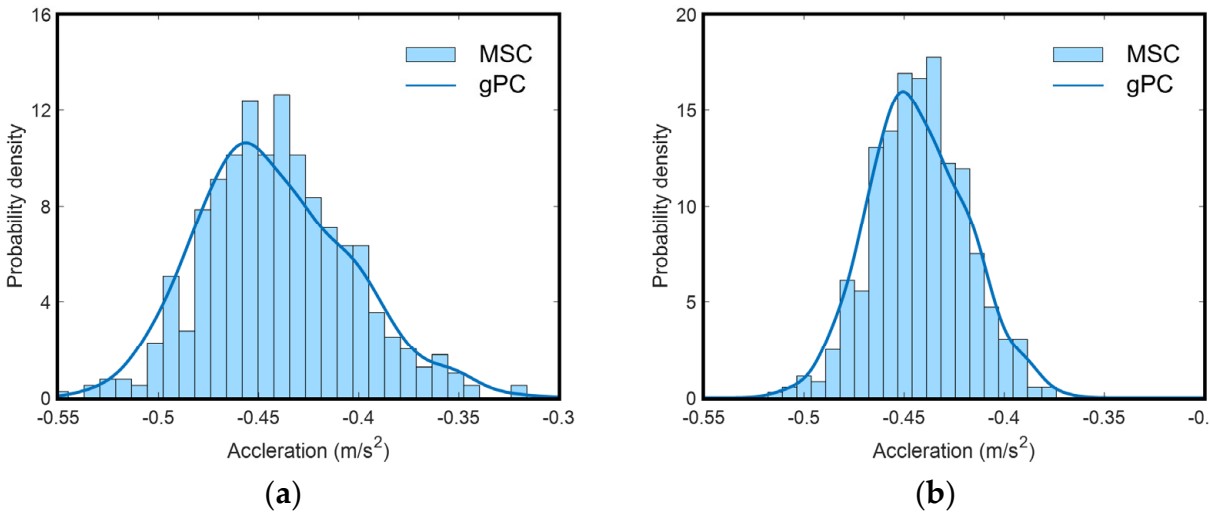

(**a**)  (**b**)

**Figure 9.** Probability density functions of the vertical accelerations when $t$ = 21.1 s at hip joint in C-class road for a different uncertain parameter, (**a**) $k_1$ and (**b**) $k_1^T$. Where shaded blocks and solid line indicate the MCS and gPC results, respectively.

Further, we used BS 6841 to quantify subjective ride comfort as a ride index [37]. To calculate the ride index, the acceleration for each axis of the human model is taken from the human–vehicle simulation, as illustrated in Figure 10. Applying Fourier transform, frequency weighting can be imposed using a convolution integral. The inverse Fourier transform of the frequency-weighted acceleration produces the time history of the acceleration, which can be weighted again by multiplying the factor at each axis. Now the root mean square of the time-weighted acceleration on each axis represents the ride index of each axis. Since the developed human model considers two-dimensional motion, in this study, only five accelerations of vertical, horizontal, and rotational directions at the hip joint and vertical and horizontal directions at the back are used in computations. Summing the riding indices of each axis, we can finally quantify the overall ride index of the human model.

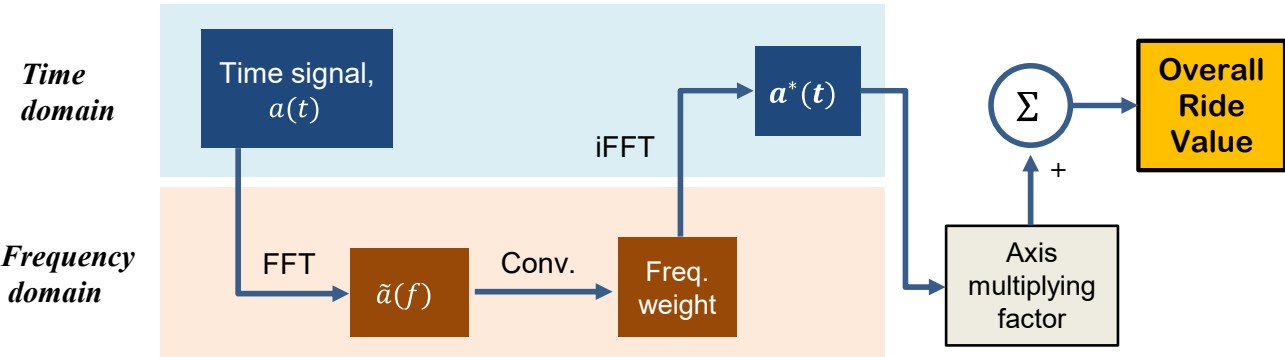

**Figure 10.** Procedure to quantify ride comfort based on BS-6841 [46].

Figure 11 shows the overall ride indices with respect to road roughness for three different uncertain parameters with the discomfort scale represented in BS 6841. We computed all ride indices and standard deviations by the approximate model based on gPC for each uncertain parameter. At each road roughness, the solid circle represents the mean of the ride indices, and the error bar represents 3σ, which implies that 99.7% of the data are in this range. The overall ride indices associated with class-A and B road roughness are placed in the 'not uncomfortable' zone, whereas those associated with class-E represent the 'fairly uncomfortable' zone. In general, class-A indicates a smooth surface, while class-C indicates average road roughness (see Table 1). Therefore, it can be predicted easily that the ride indices at E-class would place in a 'fairly uncomfortable' zone. It is also easy to understand that the standard deviation on rough road is greater than on smooth road. However, it is not easy to predict the parameter sensitive to ride index. Now comparing the error bar for uncertain stiffness, it is clear that the ride index is dominantly influenced by $k_1$ and $k_3$, which are the translation stiffness related with thigh and back of the human and seat. However, $k_1^T$, which is the torsional stiffness at hip joint of the human, is not sensitive to the ride index.

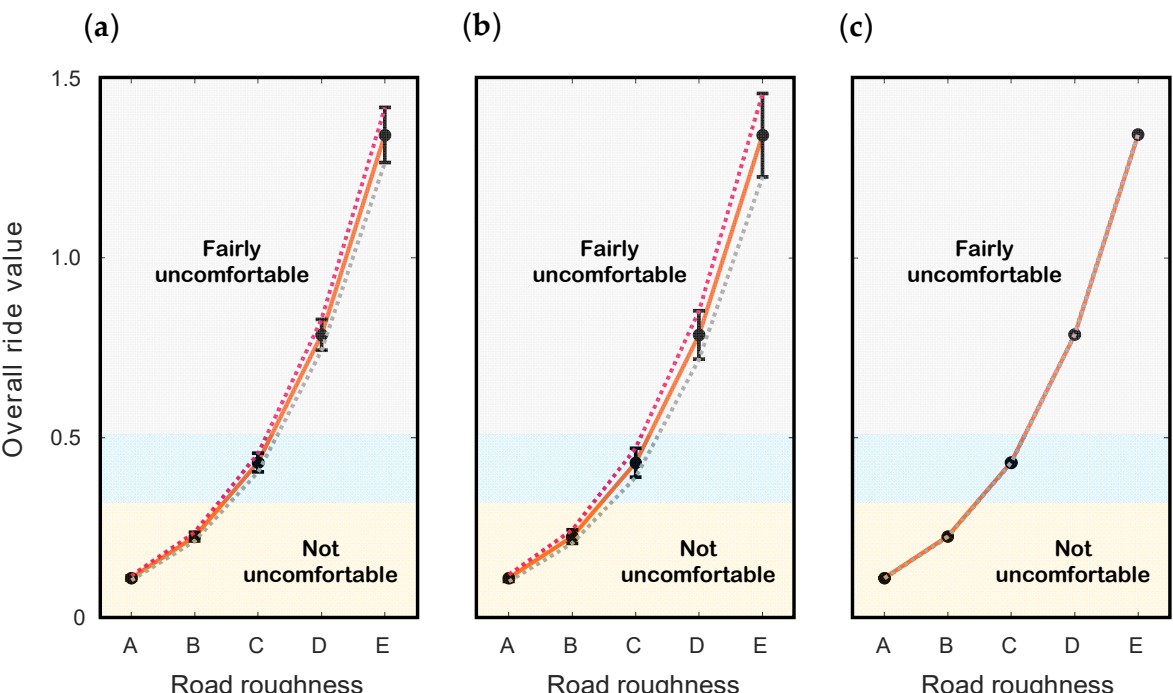

**Figure 11.** Ride indices with respect to the road roughness for a different uncertain parameter, (**a**) $k_1$ (**b**) $k_3$ and (**c**) $k_1^T$, where the error bar represents 3σ.

### 3.2. Sensitivity Analysis of Ride Comfort

To improve ride comfort, it is important to understand which segment of the vehicle or human body is closely related or sensitive to ride comfort. In the presence of uncertainties of a parameter, we performed human–vehicle simulation and obtained the probability distributions of ride index as illustrated in Figure 12. The results for translational parameters ($k$ and $c$) are displayed in the left column, and those for torsional parameters ($k^T$ and $c^T$) are in the right column. The simulations were performed when the standard deviation of each uncertain parameter has 5% of its mean value. As it may be expected, the shape of distributions is quite different depending on the uncertain parameters. Surprisingly, the range of the ride index for the translational parameters is at least 20 times wider than those for the torsional parameters. This implies that the translational parameters have much stronger effects on the uncertainty of the ride index than the torsional parameters. It can also be interpreted as the ride index is dominated by the fundamental frequency, which is mainly influenced by the translation parameter of human model [38]. In addition, it can

be seen that the distribution for $k_2^T$ and $c_2^T$ have a much shaper shape than $k_1^T$ and $c_1^T$, as shown in Figure 12b,d and Table 3. In other words, the hip joint is more sensitive than the neck part of human body. In a physical sense, it is easy to verify because the acceleration of the hip joint and trunk is more dominant than others in evaluating ride comfort [46]. In conclusion, it was found that the variations in $k_2^T$ and $c_2^T$, which connects the trunk and the head, has little effect on the movement of the hip joint and the trunk.

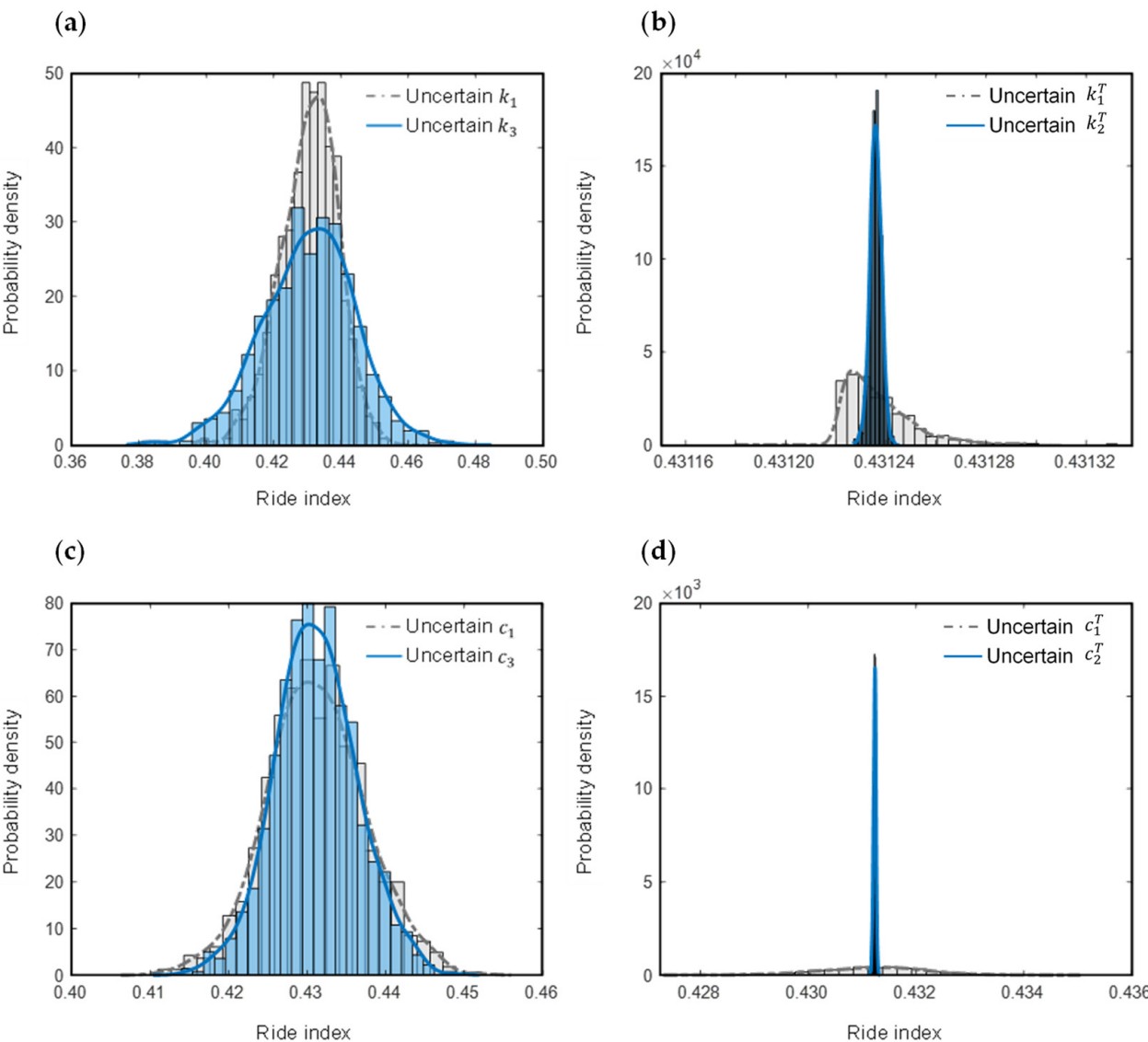

**Figure 12.** Probability density of ride index in C-class road for uncertain parameters (**a**) translational stiffness, $k_1$ and $k_3$, (**b**) torsional stiffness, $k_1^T$ and $k_2^T$, (**c**) translational damping, $c_1$ and $c_3$, (**d**) torsional damping, $c_1^T$ and $c_2^T$.

**Table 3.** Stochastic properties of ride index for the case of Figure 12.

| Stiffness | $k_1$ | $k_3$ | $k_1^T$ | $k_2^T$ |
|---|---|---|---|---|
| Mean | 0.4306 | 0.4304 | 0.4312 | 0.4312 |
| Std | $0.85 \times 10^{-3}$ | $1.36 \times 10^{-2}$ | $1.45 \times 10^{-5}$ | $2.27 \times 10^{-6}$ |
| **Damping** | $c_1$ | $c_3$ | $c_1^T$ | $c_2^T$ |
| Mean | 0.4311 | 0.4311 | 0.4312 | 0.4312 |
| Std | $0.62 \times 10^{-2}$ | $0.52 \times 10^{-2}$ | $0.97 \times 10^{-5}$ | $2.39 \times 10^{-6}$ |

It is also interesting to investigate the differences between the linear and nonlinear human models concerning human–vehicle simulation in terms of stiffness uncertainties. In this study, linearization was performed using Taylor expansion under a small motion assumption. The mean and standard deviation of the ride index are presented in Figure 13 at the top and bottom for the nonlinear and linear models, respectively. The ride indices were plotted as colormaps to simultaneously look over the effect of the uncertain parameters. For comparison, each parameter varied from 20% to 180% of the reference value and was expressed as a dimensionless value denoted by a superscript *. For example, $k_1^* = 1$ means an initial value, and $k_1^* = 0.2$ means the 20% of the initial value. The standard deviation of each uncertain parameter was also set to 5% of the mean value, and the human–vehicle simulation was performed on a C-class road. In conclusion, the mean values of the ride index for the linear and nonlinear models show similar trends. The higher the stiffness, the worse the ride comfort. However, the mean of the ride index of the nonlinear model shows a higher value than that of the linear model, and it is probably because the nonlinear model allows for large movements. For example, when $k_1^*$ and $k_3^*$ increases by 40%, the ride index increases by 25.4% from 0.451 to 0.604 in the nonlinear model (Figure 13A), while it increases by 20.5% from 0.421 to 0.529 in the linear model (Figure 13B). On the contrary, for damping, the higher the damping, the better the ride comfort. When $c_1^*$ and $c_3^*$ decreases by 40%, the ride index increases by 21.4% from 0.451 to 0.574 in the nonlinear model (Figure 14A), while it increases by 18.6% from 0.421 to 0.517 in the linear model (Figure 14B).

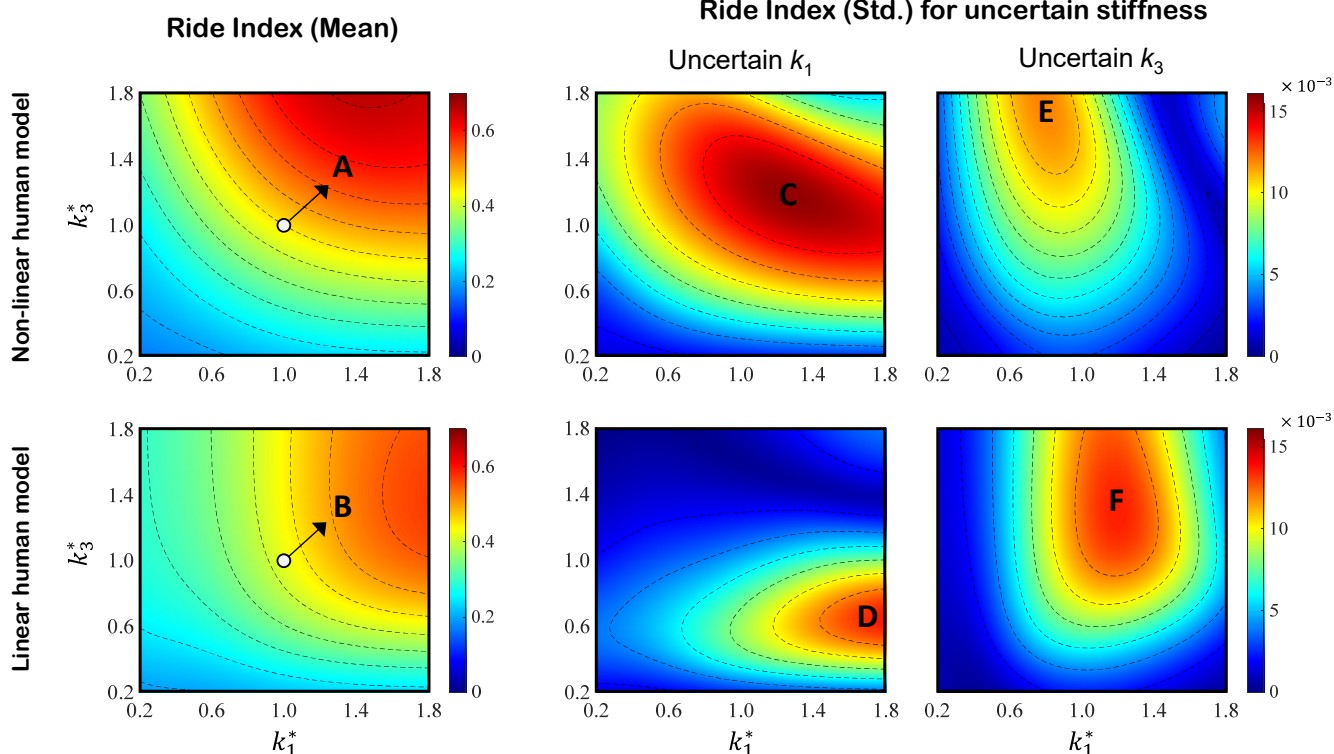

**Figure 13.** Mean and standard deviation of ride index with respect to uncertain stiffness calculated using the nonlinear and linear human models, where the 1st column is the mean, the 2nd column is the standard deviation for $k_1^*$, and the 3rd column is the standard deviation for , and the 3rd column is the standard deviation for , and the 3rd column is the standard deviation for $k_3^*$.

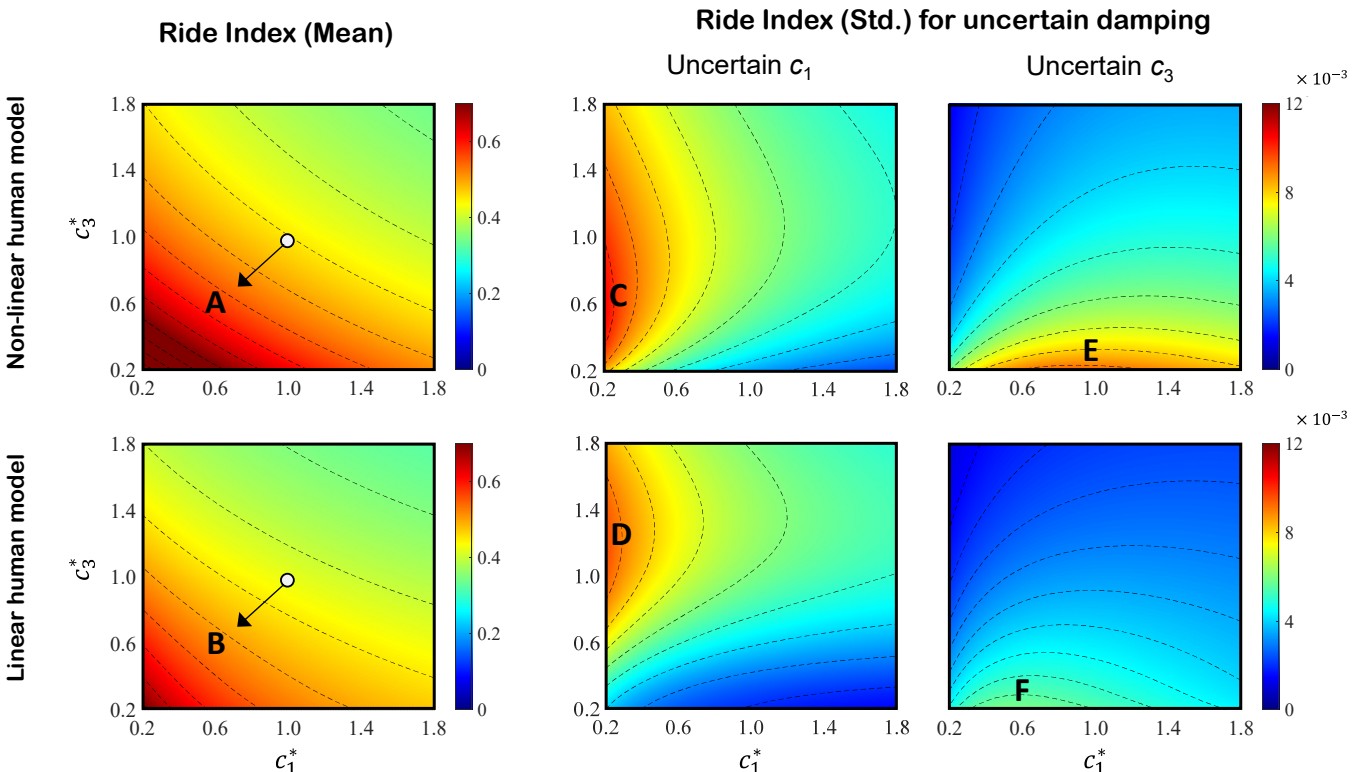

**Figure 14.** Mean and standard deviation of ride index with respect to uncertain damping calculated using the nonlinear and linear human models, where the 1st column is the mean, the 2nd column is the standard deviation for $c_1^*$, and the 3rd column is the standard deviation for $c_3^*$.

When the stiffness parameter is uncertain, the standard deviation of the ride index tends to be different depending on the presence of nonlinearity in the human body model. Interestingly, when there is uncertainty only in $k_1$, the maximums of the standard deviation exist at different locations (compare **C** and **D** in Figure 13). As mentioned earlier, the higher the standard deviations, the greater the sensitivity, and the results imply that the sensitive ranges of the uncertain parameters are significantly different. Similar results also occur in terms of $k_3$ (compare **E** and **F** in Figure 13). In the case of $c_1$ and $c_3$, the effect of standard deviations in the nonlinear and linear models are similar, but their maximums are slightly different (compare **C**, **D** and **E**, **F** in Figure 14). Based on the above results, it can be concluded that the nonlinear model responds more sensitively to the variations of the seating parameters than the linear model, leading to the need for a nonlinear model for more accurate human–vehicle simulations.

## 4. Conclusions

In the presence of parameter uncertainty, this study analyzed the ride comfort stochastically through the simulation of a fully coupled human–vehicle model using CarSim software combined with MATLAB/Simulink. Ride comfort was evaluated quantitatively as a ride index using a human–vehicle model according to the BS-6841 standard. In addition, a generalized polynomial chaos extension (gPC) framework was proposed to compute the stochastic response of the human model. Comparing the computational time of gPC with Monte Carlo simulation (MCS), the former proved to be 160 times faster than the latter.

We also carried out a parametric study of stochastic ride comfort in terms of nonlinearities and uncertain parameters in human models. Two main results were observed. First, the translational parameters of the human model are much more important than the torsional parameters when comparing the standard deviations of the ride index. Second, the use of a nonlinear human model shows that the standard deviation of the ride index is

significantly different from the linear model. Based on the above observations, it can be concluded that the uncertain quantification method and the nonlinear human model are needed for more accurate and precise ride comfort analysis.

This study also showed the distribution of ride comfort according to the road surface when there is a deviation in the stiffness and damping coefficient for drivers of average body type and weight. In a real driving environment, factors such as posture, road surface, and driving conditions are constantly changing, leading to changes in weight distribution and skin/seat deformation. As a result, even for the same driver or passenger, the response of the human body to vibration becomes unpredictable and uncertain. We expect that the proposed gPC framework can play an important role in automotive engineering as the demand for ride comfort continues to increase. For example, automotive engineers can gain insights from this study to design seats that are robust in terms of ride comfort. In this study, the parameters of $k_1$, $k_3$, $c_1$, and $c_3$ of the human model represent the springs and dampers that connect the human body to the seat cushion. Therefore, based on the results of the study, it is possible to find the optimal seat design parameters that improve ride comfort with a small standard deviation.

**Author Contributions:** Conceptualization, B.-G.S., J.-J.B. and N.K.; methodology, B.-G.S. and J.-J.B.; software, B.-G.S. and J.-J.B.; validation, B.-G.S. and J.-J.B.; visualization, B.-G.S.; writing—original draft preparation, B.-G.S. and J.-J.B.; supervision, N.K.; funding acquisition, N.K. All authors have read and agreed to the published version of the manuscript.

**Funding:** This work was supported by the National Research Foundation of Korea (NRF) grant funded by the Korean government (MIST) (No. 2020R1A2C2013256).

**Institutional Review Board Statement:** Not applicable.

**Informed Consent Statement:** Not applicable.

**Data Availability Statement:** Not applicable.

**Conflicts of Interest:** The authors declare no conflict of interest.

## Appendix A

**Table A1.** Inertial properties, geometric parameters, stiffness, and damping coefficients of the five-degrees-of-freedom human model [38].

| Parameter | Symbol | Value |
|---|---|---|
| | $m_1$ | 10.49 |
| Mass (kg) | $m_2$ | 33.98 |
| | $m_3$ | 6.67 |
| | $J_1$ | 0.23 |
| Mass moment of inertia (kgm$^2$) | $J_2$ | 2.05 |
| | $J_3$ | 0.03 |
| | $L_{tr}$ | 598.60 |
| | $L_{th}$ | 571.70 |
| | $L_h$ | 217.10 |
| | $l_1$ | 88.00 |
| Length (mm) | $l_2$ | 459.80 |
| | $l_3$ | 100.00 |
| | $l_4$ | 478.90 |
| | $T_{th}$ | 156.20 |
| | $T_{tr}$ | 224.00 |

**Table A1.** *Cont.*

| Parameter | Symbol | Value |
|---|---|---|
| Stiffness (kN/m, kNm/rad) | $k_1, k_2$ | 66.60 |
| | $k_3, k_4$ | 95.54 |
| | $k_1^T$ | 1.42 |
| | $k_2^T$ | 1.12 |
| Damping coefficient (kNs/m, kNms/rad) | $c_1, c_2$ | 0.89 |
| | $c_3, c_4$ | 0.94 |
| | $c_1^T$ | 0.30 |
| | $c_2^T$ | 0.20 |

**Appendix B**

The elements of the matrix in Equation (2) are defined as follows:

$$S_{11} = m_1 + m_2 + m_3, \quad S_{13} = -\tfrac{1}{2}m_1 L_{th}\sin\theta_1$$

$$S_{14} = -\tfrac{1}{2}L_{tr}(m_2 + 2m_3)\sin\theta_2, \quad S_{15} = -\tfrac{1}{2}m_3 L_h\ddot{\theta}_3\sin\theta_3$$

$$S_{22} = m_1 + m_2 + m_3, \quad S_{23} = \tfrac{1}{2}m_1 L_{th}\cos\theta_1$$

$$S_{24} = \tfrac{1}{2}L_{tr}(m_2 + 2m_3)\cos\theta_2, \quad S_{25} = \tfrac{1}{2}m_3 L_h\cos\theta_3$$

$$S_{33} = J_1 + \tfrac{1}{4}m_1 L_{th}^2, \quad S_{44} = J_2 + \tfrac{1}{4}m_2 L_{tr}^2 + m_3 L_{tr}^2$$

$$S_{45} = \tfrac{1}{2}m_3 L_{tr}L_h\cos(\theta_3 - \theta_2), \quad S_{55} = J_3 + \tfrac{1}{4}m_3 L_h^2$$

$$P_1 = \tfrac{1}{2}m_1 L_{th}\left(\dot{\theta}_1^2\cos\theta_1\right) + \tfrac{1}{2}L_{tr}(m_2 + 2m_3)\left(\dot{\theta}_2^2\cos\theta_2\right) + \tfrac{1}{2}m_3 L_h\left(\dot{\theta}_3^2\cos\theta_3\right)$$
$$- \left\{\left(c_3\dot{\delta}_3 + c_4\dot{\delta}_4\right) + (k_3\delta_3 + k_4\delta_4)\right\}\sin\theta_2^*$$

$$P_2 = \tfrac{1}{2}m_1 L_{th}\dot{\theta}_1^2\sin\theta_1 + \tfrac{1}{2}L_{tr}(m_2 + 2m_3)\dot{\theta}_2^2\sin\theta_2 + \tfrac{1}{2}m_3 L_h\dot{\theta}_3^2\sin\theta_3$$
$$- \left(c_1\dot{\delta}_1 + c_2\dot{\delta}_2\right)$$
$$- (k_1\delta_1 + k_2\delta_2) + \left\{\left(c_3\dot{\delta}_3 + c_4\dot{\delta}_4\right) + (k_3\delta_3 + k_4\delta_4)\right\}\cos\theta_2^*$$
$$- (m_1 + m_2 + m_3)g$$

$$P_3 = -(k_1\delta_1 + k_2\delta_2)\left(\tfrac{1}{2}T_{th}\sin\theta_1 + l_1\cos\theta_1\right)$$
$$- \left(c_1\dot{\delta}_1 + c_2\dot{\delta}_2\right)\left(\tfrac{1}{2}T_{th}\sin\theta_1 + l_2\cos\theta_1\right) + c_1^T\dot{\theta}_1^T + k_1^T\theta_1^T$$
$$- \tfrac{1}{2}m_1 g L_{th}\cos\theta_1$$

$$P_4 = \tfrac{1}{2}m_3 L_{tr}L_h\dot{\theta}_3^2\sin(\theta_3 - \theta_2)$$
$$- (k_3\delta_3 + k_4\delta_4)\left\{\tfrac{1}{2}T_{tr}\sin(\theta_2 - \theta_2^*) - l_3\cos(\theta_2 - \theta_2^*)\right\}$$
$$- \left(c_3\dot{\delta}_3 + c_4\dot{\delta}_4\right)\left\{\tfrac{1}{2}T_2\sin(\theta_2 - \theta_2^*) - l_4\cos(\theta_2 - \theta_2^*)\right\} - c_1^T\dot{\theta}_1^T + c_2^T\dot{\theta}_2^T$$
$$- k_1^T\theta_1^T$$
$$+ k_2^T\theta_2^T - \left(\tfrac{1}{2}m_2 + m_3\right)g L_{tr}\cos\theta_2$$

$$P_5 = -\tfrac{1}{2}m_3 L_{tr}L_h\dot{\theta}_2^2\sin(\theta_3 - \theta_2) - c_2^T\dot{\theta}_2^T - k_2^T\theta_2^T - \tfrac{1}{2}m_3 g L_h\cos\theta_3$$

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
