# Peer review of "Uncertainty Quantification of Ride Comfort Based on gPC Framework for a Fully Coupled Human–Vehicle Model"

_applsci, doi:10.3390/app13116785_

Round 1
Reviewer 1 Report
The title is coherent with the content of the research and focused well the main objectives of the study. The main objective of the discussion is to investigate the impact of uncertain parameters on ride comfort in a driving vehicle. A fully coupled model was developed using CarSim software and MATLAB/Simulink to simulate a driving vehicle combined with an analytical nonlinear human model. The ride index was evaluated and a framework for calculating it was proposed using the generalized polynomial method. Sensitivity analysis was performed and the results were in good agreement with Monte-Carlo simulation. The study found that the non-linearity of the human model is an important concern in the stochastic estimation of ride comfort.
The introduction is well written and easy to understand. The paper is useful from both of the theoretical and methodological point of view. Aims and methods are clearly described and supported with sufficient theoretical background. The aims of the research were fulfilled and methods of research work original and appropriate to the aims and hypothesis formulated in the introduction. This paper can be accepted with minor revision.
- The authors should use references at line 29. They can study and address the following papers (2023):
https://doi.org/10.3390/su15021528; https://doi.org/10.48295/ET.2023.91.10
- Line 54: acronym in brackets and not vice versa. Check in the manuscript similar mistakes.
- BS 6841: no reference is provide.
- The ISO 2631-1 also offer a method to measure, evaluate and assess whole-body vibration. Why the authors refer to the British Standard?
- Please assign a number for the formula presented on Page 8.
- Provide the title of the "Equation" in the text to refer to each Equation (e.g., Page 3, Line 119, etc.).
- Conclusions are lacking in terms of numerical results
The Quality of English Language is good
Reviewer 2 Report
In the presented study entitled "Uncertain Quantification of Ride Comfort based on gPC Frame-2 work for a Fully Coupled Human-vehicle Model" , the research examined" the nonlinearity of the human model is significant in evaluating stochastic ride comfort, according to the research, which looked at the effects of unknown factors on ride comfort in a driving vehicle using simulation models.
The referee carefully read the study and pointed out some concerns to the authors in order as follows.
· Each major heading should be followed by a brief descriptive phrase before the subsection. For example, between section 2 and section 2.1
· The author described the human model with the schematic diagram of the five-degree-of-freedom human given in Figure 1. But there is no detailed model for the vehicle model. This study consists of human and vehicle interaction with road excitation. For example, what is the vehicle model, quarter car model, half car model, or full car model? What is the DOF of the presented vehicle model? I strongly suggest to author for adding a schematic figure to describe to vehicle's physical model.
· The author says "Figure 6 illustrates the effects of road roughness and coupling on the reaction forces". But it is not clear what is the reaction force. Where is it occurs between which components of the vehicle?
· The authors need to discuss insightful and practical implications in the Conclusion section. The discussions must be fully stated in at least one paragraph. The greatest remark, and the one being the main reason for the necessity of revision, is the conclusion section. The conclusion of the article is more a summary of the article than a conclusion. It is a repetition of the statements and facts are already given previously. No real conclusions were drawn from the study, and no suggestions for the practical use of the results. Therefore, the conclusion section should be totally rewritten to point out the reasons for stated superiority and indicate practical advantages and use.
· The introduction section is very poor. The opening to the post has drawn a lot of attention since it doesn't include enough sources to back up the assertions and points made. The introduction's inability to clearly express the research's uniqueness or originality is still another significant problem. The study's contribution to the body of knowledge or its original viewpoint on the topic should be made clear in the introduction. The uniqueness of the writers' work should also be made very obvious, with an explanation of how it fills a gap in the existing literature or offers a new angle on the subject. The post may build a stronger foundation and more effectively engage its audience by resolving these problems.
Moghimi, H., & Ronagh, H. R. (2008). Development of a numerical model for bridge–vehicle interaction and human response to traffic-induced vibration. Engineering Structures, 30(12), 3808–3819. https://doi.org/https://doi.org/10.1016/j.engstruct.2008.06.015
Koç, M. A. M. A., & Esen, İ. (2017). Modelling and analysis of vehicle-structure-road coupled interaction considering structural flexibility , vehicle parameters and road roughness †. Journal of Mechanical Science and Technology, 31(5), 1–18. https://doi.org/10.1007/s12206-017-0913-y
Koç, M. A. (2022). A new expert system for active vibration control ( AVC ) for high ‑ speed train moving on a flexible structure and PID optimization using MOGA and NSGA ‑ II algorithms. Journal of the Brazilian Society of Mechanical Sciences and Engineering. https://doi.org/10.1007/s40430-022-03441-x
Dear editor,
My revision report has been reported to the authors and necessary arrangements are required.
Round 2
Reviewer 2 Report
The paper has been well-revised.